# Clinical and Instrumental Evaluation of Vestibular Function Before and After Cochlear Implantation in Adults

**DOI:** 10.3390/audiolres15030071

**Published:** 2025-06-15

**Authors:** Pasqualina Maria Picciotti, Tiziana Di Cesare, Daniela Rodolico, Walter Di Nardo, Jacopo Galli

**Affiliations:** 1Complex Operational Unit of Ear Nose and Throat—Department of Neuroscience, Sense Organs and Thorax, Foundation Polyclinic University A. Gemelli IRCCS, 00168 Rome, Italy; pasqualinamaria.picciotti@policlinicogemelli.it (P.M.P.); walter.dinardo@policlinicogemelli.it (W.D.N.); jacopo.galli@policlinicogemelli.it (J.G.); 2Department of Neuroscience, Sense Organs and Thorax—Catholic University of the Sacred Heart, 00168 Rome, Italy; drodolico95@gmail.com; 3Audiology and ENT, Institute for Maternal and Child Health—IRCCS “Burlo Garofolo”, 34137 Trieste, Italy

**Keywords:** cochlear implant, vestibular damage, head impulse test, caloric test, dynamic posturography, vestibular impairment after cochlear implant

## Abstract

Background/Objectives: Vestibular dysfunction is one of the main complications after cochlear implant (CI) surgery, and there are currently no standardized protocols for vestibular assessment in CI candidates. Our objectives were to investigate the incidence of vestibular impairment after CI surgery, anamnestic (age, known systemic pathologies and cause of deafness) and surgical (intraoperative complications, malposition of the CI) risk factors, and the role of vestibular assessment in the selection of the suitable ear for implantation. Methods: We included 68 adult patients (80 ears) affected by moderate-to-profound SNHL undergoing CI. The dizziness handicap inventory (DHI), the video head impulse test (VHIT), the caloric test, and dynamic posturography (DP) were used to study the vestibular function and balance before and one month after CI. The DHI was also administered 24 h after surgery. Results: Despite significative impairment 24 h after surgery (29.6 ± 30), the mean DHI score returned to preoperative values (17.9 ± 26) after one month. Dizziness persisted in case of age ≥ 65 years old, surgical difficulties, simultaneous bilateral CI, Meniere’s disease and otosclerosis, comorbidities ≥ 3, anxiety/depression, and neurological diseases. The VHIT significantly worsened in 25% of ears, while the caloric test SPV nystagmus significantly decreased in 30% of ears. In cases of preoperative unilateral weakness, the implantation of the better ear was significantly related to higher DHI scores. Only 4/68 patients had a significant persistent reduction in the postural composite score after surgery, with an increased risk of falls. Conclusions: Medical history and vestibular assessment predict the risk of vestibular damage and help to choose the CI’s side and to manage vertigo after surgery.

## 1. Introduction

Cochlear implant (CI) is considered a reliable way to restore hearing in patients with severe-to-profound sensorineural hearing loss (SNHL). Since its introduction, CI technology has significantly evolved and surgeries are more frequently being performed without a significant change in the risk of perioperative complications [1]. Despite that, vertigo and dizziness are common minor complications after cochlear implantation [2].

In fact, hearing and balance have a close anatomical and physiological relationship: on one hand, the same pathologies that involve the auditory system often also affect the vestibular system; on the other hand, the anatomical contiguity between the two systems means that the functional hearing surgery could often have significant consequences for vestibular function. The vestibular system plays a fundamental role in our daily life as it helps to maintain balance and posture. It also contributes to spatial awareness and motion perception. Although increasingly accurate testing of the vestibular pathway has become more available in the clinic, there are currently no standardized protocols for vestibular assessment in CI candidates and no consensus on which vestibular test should be conducted routinely in all patients or in relation to individual factors (age, clinical history, pre-existing balance disorders). The choice of tests differs between healthcare institutions [3]. Advanced instrumental tests could be used, as vestibular and balance screening methods based on self-assessment questionnaires, low-tech equipment, and bedside evaluation have been suggested, especially in developing countries with low resources [4].

Vestibular impairment is estimated to affect around 50% of patients with SNHL, so bilateral or unilateral vestibular dysfunction may be found in CI candidates before surgery [5]. Inner ear diseases often involve both the anterior and posterior labyrinths such as Meniere’s disease, inner ear malformations (enlarged vestibular aqueduct, incomplete partition of the cochlea type I and II) [6], congenital CMV infection [7], meningitis [8], autoimmune pathologies [9], and hereditary syndromic deafness [10].

Despite the improvements in CI surgical techniques, various complications are possible and vestibular dysfunction is still the most frequent. Technical factors such as electrode array design, surgery, and the CI model used could influence the vestibular outcomes [3]. The exact incidence of balance disorders and vestibular dysfunction in patients undergoing CI is currently unclear and controversial. Some studies indicate minimal impact on vestibular function, while others show persistent vestibular damage. Some authors even describe the improvement of body balance after CI [11], while others suggest that CI has no effect on balance [12]. The discrepant results may be due to the inadequate methodology, small sample sizes, and different vestibular assessment techniques used [13].

A recent review by Swain [14] describes that the incidence of dizziness after implantation ranges from 0.33% to 75% of cases. This great variability also emerged when considering the instrumental evaluation of vestibular function after CI: in several articles, the authors used different diagnostic methods and different timings, making difficult to perform the comparison of results. In the meta-analysis conducted by Vaz et al. in 2022 [15], the included studies used different instrumental tests (posturography, VEMPs, the VHIT, and the caloric test). Furthermore, it should be noted that some studies lack preoperative vestibular assessment, providing only postoperative data. Other confounding factors are linked to the etiology of SNHL: some anatomical conditions can increase surgical difficulty and consequent trauma [9,15,16]. We should then remember that dizziness is a very common symptom after any kind of surgery (anesthesiologic factors), especially in older patients (orthostatic hypotension [4]), and that it could be present in the absence of real vestibular dysfunction.

Different mechanisms of vestibular damage after CI surgery can be hypothesized [17]: direct trauma to the labyrinth during electrode insertion, lymphatic leak, endolymphatic hydrops, intraoperative perilymph fistula, inflammation with labyrinthitis due to foreign body reaction [18], otolith displacement [19], and vestibular electrical stimulation [20].

The aim of our study was to evaluate vestibular function before and after CI in order to clarify the impact of CI surgery on the vestibular system and the role of vestibular assessment in the selection of the suitable ear for implantation (in case of unilateral CI).

The second objective was to identify risk factors of postoperative vestibular damage.

## 2. Materials and Methods

### 2.1. Study Sample

We performed a monocentric case–crossover prospective observational study, as every patient (case) also represents their own control, because each of them was tested before and after surgery to compare results. This study was approved by the Ethical Committee of Foundation Polyclinic University A. Gemelli IRCCS, Rome, Italy (0012424/23). All patients received complete and comprehensible information about the tests administered and gave their written consent to their execution, in agreement with the ethical standards of the Declaration of Helsinki.

The inclusion criteria were as follows: adults (>18 years) affected by moderate-to-profound SNHL eligible for unilateral or bilateral CI. Individuals with a history of vestibular schwannoma (because of the involvement of the vestibular nerve and the progression of vestibular dysfunction), active middle ear disease, and congenital malformations of the auditory/vestibular system (due to potential surgical difficulty) were excluded from this study.

We included 68 patients (38 F, 30 M; mean age 50.1 ± 15.9, range 18–84 years): 57 (83.8%) were affected by bilateral severe-profound HL, 9 (13.2%) by asymmetric deafness, and 2 (2.9%) by single-sided deafness (SSD).

Causes of deafness were as follows: genetic in 13 patients (19.1%) (9 were related to connexin26 mutation and 4 were syndromic—1 MELAS, 1 Cogan, 1 Epstein, 1 Waardenburg); traumatic in 4 (5.8%) (1 electrocution and 3 temporal bone fractures); 2 (2.8%) were due to meningitis; 4 (5.8%) were due to otosclerosis; 1 (1.4%) was due to chronic cholesteatomatous otitis media; 3 (4.4%) were due to drug ototoxicity; 1 (1.4%) was due to autoimmune cerebellar ataxia; 3 (4.4%) were due to Meniere’s disease; 7 (10.3%) were due to viral infection (2 Citomegalovirus; 1 Paramyxovirus; 4 Herpes Zoster); 7 (10.3%) were due to sudden sensorineural hearing loss; 23 (33.8%) were idiopathic.

The total sample suffered from, on average, 1.4 comorbidities (range 0–6). In total, 25 (36.7%) patients were affected by hypertension, 6 (8.8%) by ischemic heart disease, 12 (17.6%) by anxious–depressive syndromes, 9 (13.2%) by autoimmune disease, 5 (7.3%) by osteoporosis, 7 (10.3%) by thyroid disease, 12 (17.6%) by diabetes, 10 (14.7%) by epilepsy or other neurological diseases, 3 (4.4%) by oncological disease, and 16 (23.5%) by other nonspecific diseases.

### 2.2. Cochlear Implantation

The round window surgical procedure was performed by senior CI surgeons. In cases of ossification of the round window, the cochleostomy technique was performed.

We performed 80 cochlear implantations: 12 (17.6%) were bilateral (7 sequential and 5 simultaneous) and 56 (82.4%) were unilateral, giving a total of 75 surgeries performed with a complete vestibular assessment before and after surgery. Implanted devices were 14 (17.5%) Advanced Bionics devices, 53 (66.2%) Cochlear devices, and 13 (16.2%) Med-el devices. “Lateral wall” electrodes were used in 30/80 ears (38.2%), while perimodiolar electrode were used in 50/80 (62.5%).

In 23 cases (28.7%), the surgery was considered “difficult” because it required longer surgical times. This was due to the perforation of the tympanic membrane needing repair in 2 cases; in 15/80 (18.7%), an ossified or poorly visible round window; and in 6/80 (7.5%), fibrosis or ossification of the cochlear duct with difficult insertion. Finally, we reported surgical complications in 4/80 cases (5%) for the following reasons: partial insertion due to fibrosis (3/80–3.75%) and the loop of the CI in the basal turn (1/80–1.25%).

### 2.3. Vestibular Evaluation

In order to evaluate quality of life in relation to vestibular disorders, we used the dizziness handicap inventory (DHI) (Italian version by Nola et al.) [21]. The questionnaire consists of 25 questions providing a choice of 3 answers, “yes” (4 points), “sometimes” (2 points) and “no” (0 points), obtaining a total score from 0 (no handicaps) to 100 (the greatest ailment imaginable). Values are considered normal (<10), borderline (10–16), mild (18–34), moderate (36–52), or severe (≥54) [22]. It was performed 24–48 h before surgery (DHI0), 24–48 h after surgery (DHI1), and one month after surgery (before CI activation) (DHI2).

In the context of cochlear implants and vestibular function, MCID (Minimal Clinically Important Difference) refers to the smallest change in the DHI (dizziness handicap inventory) scores that patients perceive as meaningful and clinically important. Studies have found that a 4-point change in DHI score is often used as the MCID, meaning that a change of 4 points or more in the DHI score is considered clinically significant.

For the purpose of this study, the authors decided to consider as the MCID any value ≥ 10% higher compared to the starting score (for example, if a patient had a total preoperative DHI score of 50, any increment ≥ 5 points was considered significant). The choice to consider the MCDI as a percentage value and not as a fixed numerical value was dictated by the desire of the authors of the present manuscript to adapt the significance of the worsening score to the starting value of each patient.

Vestibular assessment was performed 24–48 h before surgery and one month after surgery (before CI activation), including the following:-Clinical examination to assess the presence of spontaneous and/or positional nystagmus, the positivity of head-shaking test (HST), and the clinical head impulse test (HIT).-Video head impulse test (VHIT) using a VOG device (ICS Impulse, GN Otometrics) able to measure the gain of the VOR (vestibular–oculomotor reflex) in both sides. We evaluated only the horizontal canals with the patient sitting upright and by fixating a visual target in front of them. Clinicians standing behind the patient generated head impulses by moving it abruptly and unpredictably in the horizontal plane. VOR gain was automatically calculated by the system as the ratio of head to eye velocity. Based on our outpatient experience and on literature data [22,23], as per the recommendation of the manufacturer, we considered normal scores to be a VOR gain > 0.8 for each side and gain asymmetry between the two sides < 20%.-Bitermic caloric stimulation by the Fitzgerald–Hallpike technique (ICS Aircal Air Caloric Sprinkler Otometrics). It was performed in a conventional manner (air-flow of 0.8 L/min at temperatures of 50 °C and 24 °C for 60 s, in the dark, in supine position with the head raised at 30°). The nystagmus amplitude was calculated by the system as a slow phase velocity (SPV) and measured in °/s. Jonkee’s formula was used to quantify the asymmetry between the sides. Results were expressed as unilateral weakness (UW—normal < 15%) and directional preponderance degree (DP—normal < 15%) [22,23]. Bilateral hyporeflexia was calculated by SPV < 10 °/s.-Computed dynamic posturography (CDP) (Equitest, Neurocom Int. Inc., Clackamas, OR, USA) was performed with the patient standing on a dual footplate enclosed by a visual surround [20]. We used the Sensory Organization Test (SOT) with six balance conditions (eyes open/closed; visual surround steady/rotated; platform steady/rotated). Data obtained included the composite equilibrium score (CES), showing the weighted average of the different conditions, and sensory analysis (SA), showing the contribution of the different sensorial afferences (somatosensory, visual, vestibular and visual-preference). We considered a normal CES and SA to be >70 [24,25].

### 2.4. Statistical Analysis

The sample was described based on its clinical and demographic characteristics using descriptive statistics techniques. Continuous values with normal distributions, such as VOR gain and the percentage of asymmetry on the VHIT test, were expressed as mean ± standard deviation (SD). Qualitative variables were expressed as absolute and relative percentages. The Kolmogorov–Smirnov test was used to demonstrate that data were normally distributed.

The first endpoint was used to calculate the incidence of post-surgical vestibular dysfunction, considering the following as abnormal: a DHI total score greater than 10/percentage of VOR asymmetry equal or greater than 20%; VOR gain < 0.8 on the implanted side/UW equal to or greater than 15%/DP, equal to or greater than 15%/CES, and with a SA greater than 70). It was achieved by calculating for each patient the DHI score, the percentage of asymmetry in VOR gain between the implanted and non-implanted ear, the UW and DP of the caloric test, and the average scores of the two trials in the total sample. Student’s *t* test was used to compare the mean scores of the DHI questionnaire, VHIT, caloric test, and SOT before and after surgery in the total sample.

The second endpoint was to identify anamnestic (age, pathologies and cause of deafness) and surgical (surgical complications, malposition of the CI) risk factors for postoperative vestibular damage. We considered significant a worsening ≥ 10% for each score (the DHI, VOR gain on the implanted side and VOR asymmetry, SPV nystagmus on the implanted side, CES, and SA) after surgery. This issue was assessed through the χ2 test and odds ratios (ORs) with 95% confidence intervals. We considered the results statistically significant for *p* < 0.05.

## 3. Results

### 3.1. DHI

At the baseline mean, DHI0 was 17.9 ± 26 (range 0–96) and 30/75 patients (40%) showed abnormal values (≥10): 7/75–9.3% borderline (10–16), 11/75–14.6% mild (18–34), 3/75–4% moderate (36–52), and 9/75–12% severe (≥54).

At the first evaluation after surgery, the mean value of the DHI (DHI1) was 29.6 ± 30, with a significant increase (*p* = 0.014).

The analysis of the different clinical factors (Table 1) showed a significant difference between patients affected by ≥3 comorbidities (DHI1 48.3 ± 36) and those affected by <2 comorbidities (DHI1 22 ± 27) (*p* = 0.0054), with a risk of experiencing an increment ≥ 10% in DHI1 compared to DHI0 (OR = 1.2; CI = 0.8–3.4; *p* = 0.032), in particular for anxiety and depression (OR = 1.14; CI = 0.9–4.3; *p* < 0.0012) and neurological disorders (OR = 1.3; CI = 0.95–5.3; *p* = 0.005) (Table 1). Moreover, patients with recent deafness (<2 years) (post-traumatic, SSHL, and post-meningitis forms) showed significantly higher DHI1 (*p* = 0.006) scores compared to the inveterate forms. We also found a significant difference between patients who underwent unilateral CI (DHI1 24.7 ± 29.7) and patients who underwent simultaneous bilateral implantation (DHI1 66.8 ± 31.4) (OR = 3.1; CI = 0.75–20; *p* = 0.043). On the contrary, there was no significant risk in patients undergoing a second sequential CI (*p* > 0.05). Statistical analysis also demonstrated that some causes of deafness represent a risk factor for the worsening of the DHI, such as Meniere’s disease (OR = 4.1; CI = 0.83–48; *p* = 0.01) and otosclerosis (OR = 2.03; CI = 0.89–15.4; *p* = 0.039).

Analysis of age showed that the mean DHI1 was 31.1 ± 30 in patients over 65 and 25.5 ± 20 in under 65 without significant differences (*p* > 0.05); nevertheless, an age ≥ 65 years old was demonstrated to be a risk factor for the worsening of dizziness (OR = 1.4; CI = 0.98–4.2; *p* = 0.022).

Despite DHI1 being higher in difficult cases (ossification of the round window, cochleostomy, partial introduction of the CI for cochlear ossification) (DHI1 37.5 ± 36) compared to other cases (DHI1 24 ± 27), the difference was not significant (*p* > 0.05). Similarly, we found no differences based on the type of device implanted (perimodiolar electrode vs. lateral-wall electrode) (*p* > 0.05) and the surgical technique used (round window vs. cochleostomy) (*p* > 0.05).

The mean DHI2 value (1 month after surgery) was 19.9 ± 26, without significant differences compared with DHI0 (*p* > 0.05). Patients showed a decrease in the total score equal to 10.1% compared to the immediate postoperative period. Compared to DHI1, 40/75 subjects (53.3%) had a significant improvement (on average 18%) in their score, 28/75 (37.3%) had a stable total score, and 7/75 (9.3%) had a worsening score (on average of 20%). A total of 21/75 implanted patients (28%) had a worsening score compared to DHI0 (35.2 ± 32). In 14/75 cases (18.6%), the DHI2 was at least 10% higher than the preoperative score; the persistence of dizziness after 1 month was related to an age ≥ 65 years (OR = 2.5; CI = 0.7–8.5; *p* = 0.039), surgical difficulties inserting the electrode (OR = 1.3; CI = 0.99–4.5; *p* = 0.039), simultaneous bilateral CI (OR = 3.2; CI = 0.78–21; *p* = 0.005), Meniere’s disease (OR = 2.2; CI = 0.94–27; *p* = 0.040) and otosclerosis (OR = 16; CI = 1.5–172; *p* = 0.001), comorbidities ≥ 3 (OR = 2.05; CI = 0.97–6.6; *p* = 0.049), anxiety and depression (OR = 1.8; CI = 0.1–7.9; *p* < 0.001), or neurological diseases (OR = 2.1; CI = 0.98–9.4; *p* < 0.001).

### 3.2. VHIT

The mean preoperative VOR gain on the CI side was 0.91 ± 0.37 and 0.88 ± 0.34 after 1 month (*p* > 0.05). In 19/80 implanted ears (23.7%), we showed preoperative abnormal VOR gain (<0.8), 15 of which occurred bilaterally (bilateral hyporeflexia). In 34/80 ears (42.5%), we verified a worsening of the gain on the implanted side after surgery; in 20/80 ears (25%), the worsening was significant (≥0.1). The age of patients being ≥65 years was a risk for a significant reduction in gain on the implanted side (OR = 2; CI = 0.77–6.1; *p* = 0.03), as were neurological diseases (OR = 1.2; CI = 0.88–5.3; *p* = 0.048) and Meniere’s disease (OR = 6.2; CI = 1.2–72; *p* < 0.0012).

The mean gain symmetry was 16 ± 15% before surgery and 18% ± 19% one month later (*p* > 0.05). Pathological asymmetry (≥20%) was highlighted in 21/75 (28%) before surgery and in 30/75 (40%) after CI. VOR asymmetry increased significantly (>10%) in 18 implanted ears, with the risk related to an age ≥ 65 years (OR = 3; CI = 1.9–2; *p* < 0.001) and comorbidities ≥ 3 (OR = 1.2; CI = 0.89–3.5; *p* = 0.049).

In the group of patients with asymmetry (21/75), nine were implanted on the “higher-functioning side” (the side with higher gain), four bilaterally, and eight on the side with lower gain. By analyzing the relationship between the implanted ear, VOR asymmetry, and DHI, we showed that patients implanted on the higher-functioning side showed higher DHI1 (24.5 ± 35) and DHI2 (19.6 ± 25) compared to patients implanted on the side with lower VOR gain (DHI1 14 ± 19; DHI2 9 ± 12), but the risk of dizziness in the case of CI on the higher-functioning side was not significant (*p* > 0.05) (Figure 1). Finally, we also found a linear relationship between the VOR asymmetry percentage and DHI (Figure 2).

### 3.3. Caloric Test

Before CI, UW was 21.2 ± 25.8% (range 0–100), and after surgery, it was 29.5 ± 25.8% (range 0–100) without significant differences. After surgery, 36/80 ears (45%) had a decrease in the SPV nystagmus elicited by caloric stimulation on the implanted side; in 24/80 ears (30%), the worsening was significant (≥10%), and an age ≥ 65 years was the only risk factor (OR = 2.9; CI = 1–8.5; *p* = 0.04).

Before CI surgery, we found pathological asymmetry in 28/75 (37.3%) implantations: this issue affected 12 patients implanted on the side with higher functioning and 15 on the lower-functioning side (one of these bilaterally). Analysis of DHI in these two groups demonstrated that patients implanted on the better ear showed highest scores (DHI1 29 ± 31; DHI2 24 ± 29) than the others (DHI1 6 ± 9 and DHI2 13 ± 17, *p* = 0.014 and *p* = 0.048, respectively) (Figure 3). Statistical analysis confirmed that implantation on the higher-functioning side in the caloric test was a risk for the onset of dizziness after surgery (OR = 2.5; CI = 1.1–9.3, *p* = 0.039).

Patients with preoperative caloric test asymmetry and a history of bilateral simultaneous CIs achieved the highest DHI values after surgery (DHI1 83 ± 7 and DHI2 73 ± 3, *p* < 0.001).

In 24/75 implantations (32%), we found bilateral hyporeflexia before surgery, without significant differences between DHI1 and DHI2 compared to the total sample.

In 18/75 (24%) surgeries, we found a pathological DP (mean 29.9%); in 10/18 of them, the preponderance was on the side to be implanted, and in 8/18 it was towards the contralateral side. During the statistical analysis, DP was not a risk factor for postoperative dizziness.

### 3.4. Dynamic Posturography

The preoperative results showed a mean composite score of 60.3 ± 22.4%, while sensory analysis highlighted different mean percentages: somatosensory, 88.5%; visual, 70.2%; vestibular, 45.0%; and visual-preference, 81.5%. A comparison of data before and after CI did not demonstrate significant differences for the composite score (60.3 ± 19.8%) and sensory afferences (89.2%, somatosensory; 72.5%, visual; 43.9%, vestibular; and 81.5%, visual-preference).

The Sensory Organization Test revealed pathological composite scores in 38/75 patients (50.6%) before surgery, with specific sensory analysis alterations in 8/75 subjects (10.6%) for somatosensory inputs, 16/75 (21.3%) for visual inputs, 60/75 (45%) for vestibular inputs, and 6/75 (8%) for visual-preference inputs. After surgery, a pathological composite score was detected in 42/75 (56%) of implanted patients, with a somatosensory deficit in 4/75 subjects (5.3%), visual deficit in 18/75 (24%), vestibular deficit in 55/75 (73.3%), and visual-preference deficit in 6/75 (8%). Only four (out of seventy-five) patients (5.3%) had a significant (>10%) reduction in the composite score after surgery. A significant reduction was showed in 10/75 patients (13.3%) for vestibular inputs and in 8/75 (10.6%) for visual ones. No risk factor was found to be related to the modification of dynamic posturography after CI.

## 4. Discussion

The first result of the present study concerns the quality of life related to vestibular function and balance evaluated by the DHI. At the baseline, before CI, 40% of patients reported abnormal values, with a significant impairment at the first evaluation and a partial improvement at the second observation after surgery. The impairment of quality of life after implantation was related to the age of patients, the etiology of deafness (Meniere’s disease and otosclerosis), comorbidities (above all neurological and psychiatric diseases), timing characteristics (recent hearing loss), surgical difficulties, and simultaneous bilateral implantation. Our findings are in partial disagreement with the literature stating that DHI scores can be unchanged [10], improved [9,10], or worse in a low percentage [9]. However, the time points used are different from our experience, and our data refer only to the effects of ear surgery, while hearing restoration could lead to an improvement in DHI, as recently described with hearing aids [26].

The second important finding of our study is related to vestibular function measured with instrumental tests (VHIT, caloric stimulation, and dynamic posturography) before and after CI. Each of the tests used for the quantitative analysis investigated a different structure of the vestibular system, and we observed discrepant results for the different tests used, as well described in the literature [14].

We found abnormal VHIT results (VOR gain asymmetry and/or unilateral and bilateral VOR gain impairment) in about 28% of patients before surgery. We also described a worsening of the gain on the implanted side after surgery in over 40% of implanted ears, related to age, comorbidities, and Meniere’s disease. Another interesting result is the linear relationship between the VOR asymmetry percentage and DHI (Figure 2), confirming that DHI can be considered a useful tool for the evaluation of patients undergoing CI. There is no consensus in the literature about the VHIT. In a meta-analysis by Vaz et al. [14], non-significant effects of CI surgery on the VHIT are described, while a recent review by Moreno et al. [2] and other studies by West [12] and Rasmussen [9] agree with our results. Moreover, statistical analysis did not show significant differences for the VHIT between patients implanted on the better ear and the others, demonstrating that this instrumental test alone cannot be considered useful for the choice of the ear to be implanted, as previously described [12].

On the contrary, vestibular weakness to caloric stimulation demonstrated that this test could be useful for side selection. In fact, we found that the implantation of the higher-functioning side was a risk for the onset of dizziness after surgery, as highlighted by the analysis of the DHIs with the highest scores in patients implanted on the better ear than the others (Figure 3). We also found that after surgery, 45% of implanted ears showed a worsening of the SPV nystagmus elicited by caloric stimulation, and this was strictly related to an age ≥ 65 years. Moreover, patients with preoperative bithermal caloric test asymmetry who underwent bilateral simultaneous CIs achieved the highest DHI values after surgery. According to the literature [9,14,15,27,28], our results confirm that the caloric test is useful in predicting the possible onset of symptoms after surgery and guiding the surgeon towards the choice of side if it cannot be made on the basis of the audiometric and anamnestic evaluation.

The fact that different results emerged between the VHIT and the caloric test before and after CI is not surprising, since this has already been found in other pathological conditions, as we recently described in vestibular schwannoma [22] and other authors highlighted in vestibular neuritis, Meniere’s disease, and vestibular migraine [29].

Finally, dynamic posturography demonstrated an impairment of composite scores in 50.6% before surgery and 56% after surgery (*p* > 0.05), with vestibular deficit in 45% before and 73% after cochlear implantation. Specific studies on the topic are limited in number [10], in disagreement with our data. As mentioned for DHI, the difference that emerged in our study regarding the specific sensory analysis alterations could be due to the short time point evaluation, pre-CI activation, and central vestibular compensation. In any case, the high percentage of pathologic results for the dynamic posturography should be considered, independent from the absence of significant worsening after surgery. In fact, it has been demonstrated that the Sensory Organization Test is a sensitive tool for identifying patients at high risk of recurrent falls, especially in persons aged over 65 [30]. Considering that the number of older adults who are potential candidates for CI will continue to increase with the aging of the population [31], the risk of fall should be evaluated and discussed during preoperative counseling, especially in frail elderly people.

Based on our findings and the literature data [2,3], performing vestibular evaluation in the management of patients with CI is imperative to ensure comprehensive care.

The potential vestibular effects of surgery should be well assessed and discussed with CI candidates. Knowledge of the functions of both labyrinths could be crucial in the selection of a suitable ear for implantation, especially in bilaterally symmetric profound hearing loss. Additionally, vestibular testing results can offer valuable information to the surgical team, allowing them to consider a specific electrode array or surgical technique that can mitigate the impact on the vestibular system. Pre-existing vestibular disorders or balance-related problems could anticipate the potential impact on the patient’s postoperative condition and subsequent rehabilitation needs, establishing the prognosis regarding body balance [24,25].

In our experience and according to the literature, after CI, vestibular evaluation should be useful for monitoring any changes in the vestibular function resulting from the surgical procedure, thus facilitating timely intervention in the event of balance-related problems [2]. In some cases, vestibular problems may be linked to CI settings, and vestibular tests can guide audiologists and ENTs in fine-tuning these settings to minimize balance disturbances [2,32]. Several authors show a possible evolution of vestibular function after CI, which should be documented through clinical and instrumental evaluation [8,9,14]. This is especially important for patients who have temporary vestibular problems that may improve over time. On the other hand, if vestibular disorders persist, vestibular rehabilitation programs can be undertaken, customized based on the patient’s characteristics and vestibular dysfunction [1,32]. Finally, we believe that vestibular evaluation after CI could improve the knowledge of the impact of the CI on the vestibular system, improving surgical techniques and electrode fitting technology.

This study has some limitations. First, both the caloric reflex tests and the VHIT study the function of the horizontal semicircular canal. Second, the timing of follow-up may also be a limitation, as 1 month may be insufficient to the determine final vestibular function; a longer post-implant follow-up time should be used to evaluate whether vestibular function changes (also in relation to electrical stimulation) or normalizes, and this is expected to be implemented in future studies. Third, we only considered adult patients without enrolling children, and it is possible that the rate of postoperative vestibular dysfunction may be different in younger patients.

Furthermore, prospective tests for measuring utricle or saccular function and inferior or superior VIII nerves (VEMPs) were not considered in this study but should be carried out in the future.

## 5. Conclusions

It remains to be clarified which vestibular tests are best in the evaluation of patients with CI. However, vestibular evaluation is mandatory before implantation to optimize surgical outcomes and for the management of postoperative complications.

Evaluation before CI establishes baseline vestibular function and identifies potential balance problems, aiding surgical planning. Evaluation after surgery is important because some patients may experience transient or persistent balance problems or dizziness requiring timely therapeutic medical or rehabilitative interventions.

## Figures and Tables

**Figure 1 audiolres-15-00071-f001:**
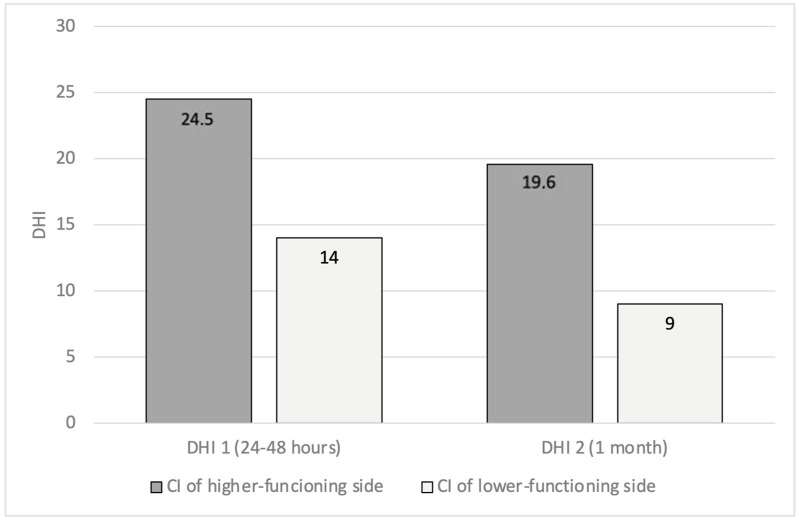
Patients with preoperative VOR asymmetry (VHIT). A comparison of DHI values, 24–48 h (DHI1) and 1 month (DHI2) after surgery, between patients implanted on the higher-functioning side and patients implanted on the other side (VOR: vestibulo-ocular reflex; VHIT: video head impulse test; DHI: dizziness handicap inventory).

**Figure 2 audiolres-15-00071-f002:**
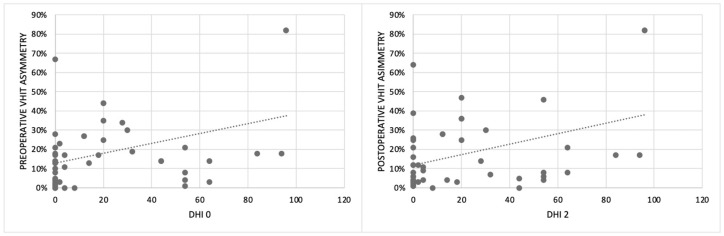
The relationship between the percentage of VOR asymmetry analyzed using the VHIT and the subjective perception of dizziness reported by patients using the DHI before surgery (DHI0) and one month after surgery (DHI2) (VOR: vestibulo-ocular reflex; VHIT: video head impulse test; DHI: dizziness handicap inventory).

**Figure 3 audiolres-15-00071-f003:**
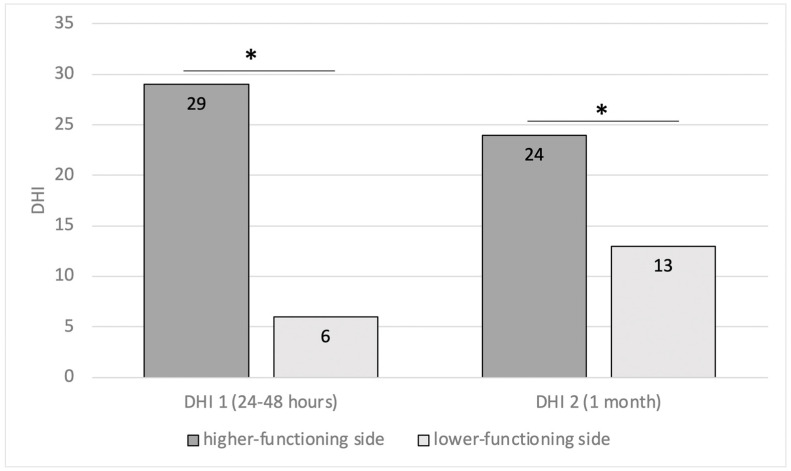
Patients with preoperative caloric asymmetry (unilateral weakness (UW)). A comparison of DHI values, 24–48 h (DHI1) and 1 month (DHI2) after surgery, between patients implanted on the higher-functioning side and patients implanted on the other side (* = *p* < 0.05) (DHI: dizziness handicap inventory).

**Table 1 audiolres-15-00071-t001:** Clinical and surgical risk factors for the DHI (dizziness handicap inventory), VHIT (video head impulse test), and caloric test worsening with odds ratios (ORs) and 95% confidence intervals (CIs).

**DHI^1^/DHI^0^ ≥ 10%** **n 25**	**Comorbidities ≥ 3** **n 27**	**Anxiety/** **depression** **n 11**	**Neurological disorder** **n 10**	**Bilateral simultaneous CI** **n 5**	**Meniere’s disease** **n 3**	**Otosclerosis** **n 4**	**Age ≥ 65** **n 20**	
	37% OR = 1.2, CI = 0.8–3.4; *p* = 0.032	36% OR = 1.14, CI = 0.9–4.3; *p* < 0.001	40% OR = 1.3, CI = 0.95–5.3; *p* = 0.005	60% OR = 3.1, CI = 0.75–20; *p* = 0.043	67% OR = 4.1, CI = 0.83–48; *p* = 0.001	50% OR = 2.03, CI 0.89–15.4; *p* = 0.039	40% OR = 1.4, CI = 0.98–4.2; *p* = 0.022	
**DHI^2^/DHI^0^ ≥ 10%** **n 14**	**Comorbidities** **≥ 3** **n 27**	**Anxiety/** **depression** **n 11**	**Neurological disorder** **n 10**	**Bilateral simultaneous CI** **n 5**	**Meniere’s disease** **n 3**	**Otosclerosis** **n 4**	**Age ≥ 65** **n 20**	**Surgical** **Difficulties** **N 23**
	26% OR = 2.05, CI = 0.97–6.6; *p* = 0.049	27% OR = 1.8, CI = 1.1–7.9; *p* < 0.001	30% OR = 2.1, CI = 0.98–9.4; *p* < 0.001	40% OR = 3.2, CI = 0.78–21; *p* = 0.005	33% OR = 2.2, CI = 0.94–27; *p* = 0.040	75% OR = 16, CI = 1.5–172; *p* = 0.001	30% OR = 2.5, CI = 0.7–8.5; *p* = 0.039	21.7% OR = 1.3, CI = 0.99–4.5; *p* = 0.008
**Postoperative VHIT gain worsening ≥ 0.1** **n 20**	**Neurological disorder** **n 10**	**Meniere’s disease** **n 3**	**Age ≥ 65** **n 20**	**Postoperative VHIT asymmetry worsening ≥ 10%** **n 18**	**Age ≥ 65** **n 20**	**Comorbidities** **≥ 3** **n 27**	**Postoperative caloric test SPV worsening ≥ 10%** **n 24**	**Age ≥ 65** **n 20**
	30% OR = 1.2, CI = 0.88–5.3; *p* = 0.048	66% OR = 6.2, CI = 1.2–72; *p* < 0.001	37% OR = 2, CI = 0.77–6.1; *p* = 0.03		40% OR = 3, CI = 1–9.2; *p* < 0.001	26% OR = 1.2, CI = 0.89–3.5; *p* = 0.049		50% OR = 2.9, CI = 1–8.5; *p* = 0.04

## Data Availability

The data presented in this study are available on request from the corresponding author.

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
