# Peer review of "Clinical and Instrumental Evaluation of Vestibular Function Before and After Cochlear Implantation in Adults"

_audiolres, 2025, doi:10.3390/audiolres15030071_

Round 1
Reviewer 1 Report
Comments and Suggestions for Authors
Dear Authors,
This is a well-designed and clinically meaningful study that offers valuable insights into the impact of cochlear implantation on vestibular function. The prospective approach, combined with both subjective and objective vestibular assessments, strengthens the relevance and reliability of your findings. Notably, your results reinforce the role of caloric asymmetry in guiding surgical side selection and highlight critical risk factors such as patient age and comorbidities.
To further enhance the scientific rigor and clinical utility of your manuscript, I recommend the following revisions:
-
Consider including a matched control group (e.g., non-implanted or healthy subjects) or referencing normative values for all vestibular tests to strengthen your comparisons and interpretations.
-
The 1-month postoperative evaluation window may not adequately reflect long-term vestibular outcomes. Please consider discussing this limitation and recommending longer follow-up in future studies.
-
While the study includes key evaluations, the absence of tests such as VEMPs (to assess utricle/saccule function) should be acknowledged as a limitation, with suggestions for inclusion in future investigations.
-
Some odds ratios are reported without corresponding confidence intervals or exact p-values. Please ensure consistency in statistical reporting.
-
Indicate whether multivariate analyses were performed to account for potential confounding variables (e.g., age, comorbidities, surgical technique).
-
Minor grammar and phrasing issues were noted (e.g., “menage” should be “manage”, “deafness aetiology” should be “etiology of deafness”). A final language edit by a native speaker or professional editor is advised to enhance clarity and fluency.
Additional Limitations to Highlight:
--The follow-up period (1 month) may not be sufficient to capture full vestibular compensation or long-term outcomes.
--Saccular/utricular testing (e.g., VEMP) and rotational chair testing were not included.
--Preoperative VOR data for the non-implanted side is not analyzed comparatively.
With these revisions, your work will offer a stronger and more comprehensive contribution to the literature on cochlear implantation and vestibular health.
Sincerely,
Reviewer
yes
Author Response
This is a well-designed and clinically meaningful study that offers valuable insights into the impact of cochlear implantation on vestibular function. The prospective approach, combined with both subjective and objective vestibular assessments, strengthens the relevance and reliability of your findings. Notably, your results reinforce the role of caloric asymmetry in guiding surgical side selection and highlight critical risk factors such as patient age and comorbidities.
To further enhance the scientific rigor and clinical utility of your manuscript, I recommend the following revisions:
- comment: Consider including a matched control group (e.g., non-implanted or healthy subjects) or referencing normative values for all vestibular tests to strengthen your comparisons and interpretations.
response: We thank the reviewer for the comment. The design of the study was not a case-control observational one but a case-crossover prospective observational study (each patient was also its own control as tests were done for each case before and after surgery).
For what concern the normative values, you know that there are no universally recognized values ​​for both VHIT and caloric test to distinguish the results as pathological or normal, but these vary from author to author. Based on our previous work (Picciotti PM, Rolesi R, Rossi G, Tizio A, Sergi B, Galli J. (2024). Caloric test, qualitative and quantitative vHIT analysis in vestibular schwannoma. Otology and Neurotology) and literature data (Molnár A, Maihoub S, Tamás L, Szirmai Á. A possible objective test to detect benign paroxysmal positional vertigo. The role of the caloric and video-head impulse tests in the diagnosis. J Otol. 2022 Jan;17(1):46-49. doi: 10.1016/j.joto.2021.11.001), we considered as indicative of vestibular damage a percentage of asymmetry equal to or greater than 20% for VHIT and unilateral weakness/directional preponderance degree equal to or greater than 15% for the caloric test.
In any case, independently from the normative values of each test, we considered significant a worsening ≥ 10% of each score after surgery (as we said in the statistical analysis section). Setting the significance of worsening >10% was a study choice. It means that we simply identified for each test the risk factors of a worsening >10% of the score. We do not have a reference for that, since it was the choice of our team (and the study objective) to calculate the risk factors significantly correlated to a worsening of at least 10% for each test.
- comment: The 1-month postoperative evaluation window may not adequately reflect long-term vestibular outcomes. Please consider discussing this limitation and recommending longer follow-up in future studies.
response: The postoperative evaluation was done one month after surgery because the tests were done immediately before the activation of the cochlear implant (we didn’t want the interference of the electrical stimulation of the cochlear implant). In any case, at the end of the discussion, we specified as limitation of the study: “Second, the timing of follow-up may also be a limitation, as 1 month may be insufficient to determine final vestibular function. A longer post-implant follow-up time should evaluate whether vestibular function changes (also in relation to electrical stimulation) or normalizes and it is expected in future studies.”
- comment: While the study includes key evaluations, the absence of tests such as VEMPs (to assess utricle/saccule function) should be acknowledged as a limitation, with suggestions for inclusion in future investigations.
response: We perfectly agree with you, in fact in the final part of the discussion we said: “Furthermore, prospective tests to measure utricle or saccular function and inferior or superior VIII nerve (VEMP) were not considered in this study but should be added in the future.”
- comment: Some odds ratios are reported without corresponding confidence intervals or exact p-values. Please ensure consistency in statistical reporting.
response: Results have been revised with odds ratios and their corresponding confidence intervals and p values. We also added a table to facilitate the interpretation of the results.
- comment: Indicate whether multivariate analyses were performed to account for potential confounding variables (e.g., age, comorbidities, surgical technique).
response: multivariate statistics were not performed because the main objectives of the study were to find how each variable (age, comorbidities, surgical technique, surgical difficulty) impacted on the risk of vestibular damage considering them independently from each other. That’s why we used univariate statistics (the χ2 test and the odds ratios (OR) with 95% confidence intervals and student t-test) considering each variable separately.
- comment: Minor grammar and phrasing issues were noted (e.g., “menage” should be “manage”, “deafness aetiology” should be “etiology of deafness”). A final language edit by a native speaker or professional editor is advised to enhance clarity and fluency.
response: The language has been revised as suggested.
Additional Limitations to Highlight:
--The follow-up period (1 month) may not be sufficient to capture full vestibular compensation or long-term outcomes.
response: The postoperative evaluation was done one month after surgery because the tests were done immediately before the activation of the cochlear implant (we didn’t want the interference of the electrical stimulation of the cochlear implant). in any case, at the end of the discussion, we specified as limitation of the study: “Second, the timing of follow-up may also be a limitation, as 1 month may be insufficient to determine final vestibular function. A longer post-implant follow-up time should evaluate whether vestibular function changes (also in relation to electrical stimulation) or normalizes and it is expected in future studies.”
--Saccular/utricular testing (e.g., VEMP) and rotational chair testing were not included.
response: We perfectly agree with you, in fact in the final part of the discussion we said: “Furthermore, prospective tests to measure utricle or saccular function and inferior or superior VIII nerve (VEMP) were not considered in this study but should be added in the future.”
--Preoperative VOR data for the non-implanted side is not analyzed comparatively.
response: VHIT (VOR) data of the non-implanted side were collected from each patient but they were not relevant for the purposes of the study, so they were not reported in the results.
With these revisions, your work will offer a stronger and more comprehensive contribution to the literature on cochlear implantation and vestibular health
Reviewer 2 Report
Comments and Suggestions for Authors
This is a prospective observational study, the authors must describe how they explained and obtained informed consent from the participants of the study to the subjects.
In the results, there are many Odds ratio, the authors should add the confidence interval and may create a table to explain this.
Author Response
comment: This is a prospective observational study, the authors must describe how they explained and obtained informed consent from the participants of the study to the subjects.
response: We thank very much reviewer 2 for his precious work and suggestions. Written consent was obtained from all patients and this important part was added in the methods section.
comment: In the results, there are many Odds ratio, the authors should add the confidence interval and may create a table to explain this.
response: The suggestions of reviewer 2 were very useful to improve the significance of our manuscript. Results have been revised with odds ratios and their corresponding confidence intervals and p values. We also added a table to facilitate the interpretation of the results, as suggested.
Reviewer 3 Report
Comments and Suggestions for Authors
Abstract: consider citing a little lit regarding why it is important to investigate vestibular function prior to CI. Line 20: does posturography examine vestibular function? Line 25: the statement about vHIT is understandable, but the 'caloric test SPV in 30% ears' is not - did this function decline? The abstract would be improved by giving us some idea of DHI at baseline, especially as you say within one month of CI this had dropped again. The comment in line 27 has relevance for falls, doesn't it? These are adults and so are at risk in any case, so consider contextualising this.
Introduction: line 36 - rather an exclusionary statement from a positivist perspective, I would have thought. Reference? Signing and Deaf completely overlooked. Please consider rewriting the last sentence of the first paragraph as it does not make sense.
Line 43 - there certainly are some recommendations, particularly for low and middle income countries which are not rich in resources e.g., Rogers, C. (2021). Perspectives: evaluation of older adult cochlear implant candidates for fall risk in a developing country setting. Frontiers in neurology, 12, 678773.
Line 62-69 - I understand the focus on specialised testing - however, you are ignoring the presence of dizziness and vertigo as symptoms and these cannot be linked to deficits (or not) on vestibular testing can they? Some discussion here would be helpful.
Materials and methods: line 81 - consider rephrasing monocentric case-cross - not sure these words will be familiar in terms of design. If you mean in a single centre, then this is better wording.
Some justification of your exclusion criteria is required (line 85).
Please rephrase line 117 - DHI does not predict presence of dizziness, does it?
Line 133 provide ref for vHIT norms.
Line 140 - 142 please provide a reference for your caloric norms - not in line with guidelines.
Line 161 - please provide a ref for the 10% worsening of score - how does this link to the responsiveness of the various tests (MCID, MCD etc)?
Use of Bonferonni given the small sample size and repeated stats? Did I miss information on how missing data were managed? How was distribution of the data checked?
Results
Line 169 - difficult with such a low DHI to know just how bad those with higher scores were - can we see a histogram maybe? And by the way what do you mean by abnormal values? Vague. Line 195 - some lack of clarity about what the DHI measures is noted again. By the way, is the translation into Italian validated and is this mentioned in the materials section?
Line 218 - prevalent side?
Line 238 - what do you mean by worsening of SPV? Less responsive? Confusing.
One of your most interesting findings is the computerised posturography being quite challenging in many patients preop - suggesting fall risk, which I hope is picked up in the discussion.
Discussion
Note my reservations about the DHI and how you are interpreting it.
Line 321 - not sure what you mean by dissociation? We know these tests measure somewhat different things and one is capable of showing some level of compensation? It could be how you are expressing it.
Line 325 - you need to make a much bigger thing of the CDP testing and its implication for daily living regarding risk of falls in this population. Don't be too tunnel visioned on vestibular function per se.
Rest of the discussion and conclusion is reasonable.
Author Response
We thank very much reviewer 3 for its comments that improved the quality of our manuscript.
comment: Abstract: consider citing a little lit regarding why it is important to investigate vestibular function prior to CI.
response: “Vestibular disfunction is one of the main complications after cochlear implant (CI) surgery and there are currently no standardized protocols for vestibular assessment in CI candidates.” The sentence has been added as background in the abstract.
comment: Line 20: does posturography examine vestibular function? response: “…and balance” has been added.
comment: Line 25: the statement about vHIT is understandable, but the 'caloric test SPV in 30% ears' is not - did this function decline? response: Yes, the function decline. It was better specified as you suggested.
comment: The abstract would be improved by giving us some idea of DHI at baseline, especially as you say within one month of CI this had dropped again. response: Mean DHI values before and after surgery have been added in the abstract.
comment: The comment in line 27 has relevance for falls, doesn't it? These are adults and so are at risk in any case, so consider contextualising this. response: The increased risk of falls has been mentioned as you suggested.
comment: Introduction: line 36 - rather an exclusionary statement from a positivist perspective, I would have thought. Reference? Signing and Deaf completely overlooked. Please consider rewriting the last sentence of the first paragraph as it does not make sense. response: We perfectly agree with you; in fact, we changed the sentence as you described.
comment: Line 43 - there certainly are some recommendations, particularly for low and middle income countries which are not rich in resources e.g., Rogers, C. (2021). Perspectives: evaluation of older adult cochlear implant candidates for fall risk in a developing country setting. Frontiers in neurology, 12, 678773. response: We thank very much reviewer 3 for sharing this interesting reference with us; it has been added to the text as suggested.
comment: Line 62-69 - I understand the focus on specialised testing - however, you are ignoring the presence of dizziness and vertigo as symptoms and these cannot be linked to deficits (or not) on vestibular testing can they? Some discussion here would be helpful.response: If I understand what you mean, you are suggesting that sometimes vertigo or dizziness could be due to other postoperative factors, and independent from a real vestibular damage. the suggestion has been added.
comment: Materials and methods: line 81 - consider rephrasing monocentric case-cross - not sure these words will be familiar in terms of design. If you mean in a single centre, then this is better wording. response: Case-cross (or case-crossover study) means that every patient (case) also represents its own control because each case was tested before and after surgery to compare results.
comment: Some justification of your exclusion criteria is required (line 85). response: It has been added: “Individuals with history of vestibular schwannoma (because of the involvement of the vestibular nerve and the progression of vestibular dysfunction), active middle ear disease and, congenital malformations of the auditory/vestibular system (due to potential increasing surgical difficulty) were excluded from the study.”
comment: Please rephrase line 117 - DHI does not predict presence of dizziness, does it? response: It was done as suggested: “In order to evaluate the quality of life in relation to vestibular disorders we used the Dizziness handicap Inventory (DHI)”
comment: Line 133 provide ref for vHIT norms. Line 140 - 142 please provide a reference for your caloric norms - not in line with guidelines.
response: “Based on our outpatient experience and on literature data [22, 23], we considered normal: VOR gain > 0.8 for each side and gain asymmetry between the two sides < 20%.”
Results were expressed as unilateral weakness (UW - normal < 15%) and directional preponderance degree (DP - normal < 15%) [22, 23].
For what concern the normative values, you know that there are no universally recognized values for both VHIT and caloric test to distinguish the results as pathological or normal, but these vary from author to author. Based on our previous work (Picciotti PM, Rolesi R, Rossi G, Tizio A, Sergi B, Galli J. (2024). Caloric test, qualitative and quantitative vHIT analysis in vestibular schwannoma. Otology and Neurotology) and literature data (Molnár A, Maihoub S, Tamás L, Szirmai Á. A possible objective test to detect benign paroxysmal positional vertigo. The role of the caloric and video-head impulse tests in the diagnosis. J Otol. 2022 Jan;17(1):46-49. doi: 10.1016/j.joto.2021.11.001), we considered as indicative of vestibular damage a percentage of asymmetry equal to or greater than 20% for VHIT and unilateral weakness/directional preponderance degree equal to or greater than 15% for the caloric test.
comment: Line 161 - please provide a ref for the 10% worsening of score - how does this link to the responsiveness of the various tests (MCID, MCD etc)?
response: Setting the significance of worsening >10% was a study choice. It means that we simply identified for each test the risk factors of a worsening >10% of the score. We do not have a reference for this, since it was the choice of our team (and the study objective) to calculate the risk factors significantly correlated to a worsening of at least 10% for each test.
comment: Use of Bonferonni given the small sample size and repeated stats? Did I miss information on how missing data were managed? How was distribution of the data checked? response: We don’t have missing data for this study because we included only patients able to do the tests before and after CI (with a complete evaluation). Kolmogorov-Smirnov Test was used to demonstrate that data were normally distributed. It has been added in the statistical analysis.
Results
comment: Line 169 - difficult with such a low DHI to know just how bad those with higher scores were - can we see a histogram maybe? response: We described the distribution of patients according to DHI values in the text: “7/75-9.3% borderlines (10-16), 11/75-14.6% mild (18-34), 3/75-4% moderate (36-52) and 9/75-12% severe (≥54).” comment: And by the way what do you mean by abnormal values? Vague. response: In the methods we specified that “Values are considered normal (<10)”. comment: Line 195 - some lack of clarity about what the DHI measures is noted again. response: “We corrected: Patients showed a decrease of the total score” comment: By the way, is the translation into Italian validated and is this mentioned in the materials section? response: Yes, it was mentioned in the methods: “we used the Dizziness handicap Inventory (DHI) (Italian version by Nola et al.) [18]”.
comment: Line 218 - prevalent side? response: We specified the term we used: “prevalent” side (side with higher gain). In any cas it was replaced with “higher-functioning side”
comment: Line 238 - what do you mean by worsening of SPV? Less responsive? Confusing. response: Yes, we mean less responsive. It was replaced with “decrease”.
comment: One of your most interesting findings is the computerised posturography being quite challenging in many patients preop - suggesting fall risk, which I hope is picked up in the discussion. response: Fall risk has been added in the discussion
Discussion
comment: Note my reservations about the DHI and how you are interpreting it. response: We thank you for your suggestion about DHI, discussion has been revised.
comment: Line 321 - not sure what you mean by dissociation? We know these tests measure somewhat different things and one is capable of showing some level of compensation? It could be how you are expressing it. response: It was replaced by “The different results emerged”
comment: Line 325 - you need to make a much bigger thing of the CDP testing and its implication for daily living regarding risk of falls in this population. Don't be too tunnel visioned on vestibular function per se. response: Discussion has been revised.
Rest of the discussion and conclusion is reasonable.
Round 2
Reviewer 3 Report
Comments and Suggestions for Authors
Thank you for answering some of the points raised. Other points have had no response, weakening the paper. Assumptions which may be challenged include how you have used and scored the DHI, lack of use of MCD and MCID data as previously requested, lack of clarity over p-values and Bonferroni correction, and perhaps most importantly, thinking that test results are linked to the experience of vestibular symptoms - when they are not necessarily. Think of PPPD as an example - often patients are very impaired yet their results on specialised testing are normal, are they not?
Comments on the Quality of English LanguageI am sure if you publish the copy editor will correct any issues.
Author Response
Thank you for answering some of the points raised. Other points have had no response, weakening the paper.
Assumptions which may be challenged include
- how you have used and scored the DHI,
response: in the methods section we described how we used and scored the DHI:” The questionnaire consists of 25 questions providing a choice of 3 answers: "yes" (4 points), "sometimes" (2 points) and "no" (0 points) obtaining a total score from 0 (no handicaps) to 100 (the greatest ailment imaginable). Values are considered normal (<10), borderlines (10-16), mild (18-34), moderate (36-52) and severe (≥54) [19]. It was performed 24-48 hours before surgery (DHI0), 24-48 hours after surgery (DHI1) and one month after surgery (before CI activation) (DHI2).
- lack of use of MCD and MCID data as previously requested,
response: “In the context of cochlear implants and vestibular function, MCID (Minimal Clinically Important Difference) refers to the smallest change in DHI (Dizziness Handicap Inventory) scores that patients perceive as meaningful and clinically important. Studies have found that a 4-point change in DHI score is often used as the MCID, meaning a change of 4 points or more in DHI score is considered clinically significant.
For the purpose of the study, the authors decided to consider as MCID any value ≥ 10% compared to the starting score (for example, if a patient had a total preoperative DHI score of 50, any increment ≥ 5 points was considered significant). The choice to consider the MCDI as a percentage value and not as a fixed numerical value was dictated by the desire of the authors of the present manuscript to adapt the significance of the worsening to the starting value of each patient.”
It was added to the methods section.
- lack of clarity over p-values and Bonferroni correction,
Response: We do not understand what you mean with unclear p-values. For each comparison between mean values performed with the t-test the statistical significance was specified in the text. The same is true for the chi-square test; the calculation of each OR was accompanied by the corresponding confidence interval and the significance, as in table 1.
-The Bonferroni correction should not have been applied in our statistical analysis since each analyzed variable (VHIT, DHI, caloric test) was correlated with each other by the same cause (i.e. post-surgical vestibular damage). In this case (variables correlated with each other) the risk of error of lowering the statistical significance with the Bonferroni correction would be higher than a possible error that the significance found was due to chance.
- and perhaps most importantly, thinking that test results are linked to the experience of vestibular symptoms - when they are not necessarily. Think of PPPD as an example - often patients are very impaired yet their results on specialised testing are normal, are they not?
response: We are sorry but all the authors of this manuscript disagree with you on this point. PPPD is a very different condition than an acute vestibular damage following oto-surgery, such as cochlear implantation. PPPD is for definition a common long-lasting perception of dizziness or vertigo, which is usually not quantifiable with instrumental tests. The measurement of the symptoms then becomes fundamental to base the diagnosis and follow-up in this condition.
The acute postsurgical vestibular damage due to round window opening, cochleostomy and electrode insertion, it is instead quantifiable through instrumental tests, and this was one of the main objectives of the study. In any case, we tried to give importance to the quantification of the impact of symptoms on patients' daily life using DHI for this purpose. The accurate evaluation of symptoms and the outcome of instrumental tests is very important for us, both for the diagnosis and for the follow-up of patients and we have carefully described the results of all the methods used (DHI questionnaire, caloric test, vhit, posturagraphy) for this purpose, to show and try to quantify the results of both aspects (instrumental and non-instrumental) and any relationship between them. As you can see in figure II for example, the VHIT asymmetry and the score of DHI were strictly related to each other.